



7 An Automated Method for Preparing and Calibrating

9 Electrochemical Concentration Cell (ECC) Ozonesondes

20 Francis. J. Schmidlin[1] and Bruno A. Hoegger[2]

1) NASA/GSFC/Wallops Flight Facility; Wallops Island, Va. 23337 (Emeritus). E-mail: francis.j.schmidlin@nasa.gov
2) Bruno Hoegger Scientific Consulting; Marly, Switzerland CH1723. E-mail: hoegger.consulting@bluewin.ch





Abstract

In contrast to the legacy manual method used to prepare, condition, and calibrate the
Electrochemical Concentration Cell (ECC) ozonesonde an automated digital calibration
bench similar to one developed by MeteoSwiss at Payerne, Switzerland was established
at  NASA's Wallops Flight Facility and provides reference measurements of the same
ozone partial pressure as measured by the ECC. The purpose of an automated system is to
condition and calibrate ECC cells before launching on a balloon. Operation of the digital
calibration bench is simple and real-time graphs and summaries are available to the
operator; all information is archived. The parameters of interest include ozone partial
pressure, airflow, temperature, background current, response, and time (real and elapsed).
ECC cells, prepared with 1.0 percent solution of potassium iodide (KI) and full buffer,
show increasing partial pressure values when compared to the reference as partial
pressures increase. Differences of approximately 5-6 percent are noted at 200 nb.
Additional tests with different concentrations revealed the Science Pump Corp (SPC) 6A
ECC with 0.5 percent KI solution and one-half buffer agreed closer to the reference than
the 1.0 percent cells, this is in agreement with results of multi-sonde comparisons
obtained during BESOS. The information gained from the automated system allows a
compilation of ECC cell characteristics, as well as calibrations. The digital calibration
bench is recommended for ECC studies as it conserves resources.



1. Introduction
Measurement disagreement between similar or identical instruments seems to be an
historical problem. Intercomparisons are generally conducted when new instruments are
introduced and when operational changes or improved procedures become available.
Such comparisons should be made under the same environmental conditions and include
a reference instrument as an aid for checking the accuracy and reliability of the
instruments. This would be ideal as a standard procedure. Unfortunately, balloon-borne
ozone reference instruments are not usually available, mostly because they are too
expensive for other than occasional use or to expend on non-recoverable balloon
packages. Ozonesonde pre-flight calibrations are conducted, however these are basically
single point calibrations made prior to its release. An automated system designed to
condition and calibrate the Electrochemical Concentration Cell (ECC) ozonesonde was
fabricated at Wallops Flight Facility. The automated system can provide calibration over
a wide range of ozone partial pressures. This system, designated the digital calibration
bench, enables consistent conditioning and calibration of the ECC along with
measurements of a reference value. In this paper the term ECC refers only to the Science
Pump Corp. (SPC) 6A ECC ozonesonde, although the automated system can
accommodate the EnSci ozonesonde as well.
There are a variety of ground-, aircraft-, satellite-, rocket-, and balloon-borne instruments
available to measure the vertical structure of atmospheric ozone and its total content.
These instruments operate on different principles of measurement (Fishman et al, 2003;
Kohmyr, 1969; Krueger, 1973; Holland et al, 1985; Hilsenrath et al, 1986; Sen et al,
1996). Although their spatial distribution is limited, balloon-borne Electrochemical
Concentration Cell (ECC) ozonesondes have had a key role as a source of truth for the
other instruments and for establishing algorithms necessary for the retrieval of satellite
observations. Manual preparation of the ECC requires hands-on contact by an operator.
Reducing subjectivity is important and was considered serious enough to engage in the
fabrication of the automated system. The user is prompted throughout the calibration





process while utilizing real-time graphs and summaries. The digital calibration bench
provides consistent preparation procedures. ECC measured ozone partial pressures vs.
reference partial pressures are discussed and the results corroborated with similar
comparison data obtained during the the 2004 comparison on the Balloon Experiment on
Standards for Ozonesondes (BESOS) mission (Deshler et al, 2008) and with dual ECC
comparisons at Wallops Island.

Notwithstanding efforts to enhance ECC performance (Smit et al, 2004, 2007, 2014; Kerr
et al, 1994;  Johnson et al, 2002; Torres, 1981) there remain uncertainties. Barnes (*1982*)
and Barnes et al (*1985*) estimated the accuracy of the ECC as 5-10 percent and also
pointed out that the accuracy varied with altitude. Uncertainties also arise from poor
compensation for the loss of pump efficiency; erroneous background current; air flow
temperature error and whether measured at the proper location; and, the use of the
appropriate potassium iodide (KI) concentration. Understanding the influence these
parameters have on the ozonesonde measurement capability is particularly important.
The digital calibration bench is able to measure these parameters in a consistent way over
a range of partial pressures.

2 Digital Calibration Bench Description and Operational Procedure

2.1 Description

The computer-controlled preparation and calibration bench fabricated at NASA Wallops
Flight Facility follows the design of a similar bench developed by MeteoSwiss scientists
B. A. Hoegger and G. Levrat at Payerne, Switzerland. The MeteoSwiss digital calibration
bench was first available in the 1990's and continues to be used and is updated
periodically. A comparable bench furnished by MeteoSwiss to the meteorological station
at Nairobi, Kenya also has been in use for a number of years. The Wallops Island ozone
site was interested in the digital bench because of its capability to provide precise and
repeatable preparation of ECC's, and its automated feature requires less interaction with
the ECC then the manual preparation method.




The Wallops digital calibration bench, shown in Fig. 1, consists of three major
components: 1) mass flow meter to control air flow, 2) an ozone generator and analyzer
(UV photometer), and 3) computer necessary to automate the timing of the programmed
functions and process the data. Another important component, the glass manifold, enables
the simultaneous distribution of the air flow to the ECC's and the UV photometer. The
manifold also is a buffer maintaining constant air flow and inhibiting flow fluctuation. A
graphical user interface controls the various input and output functions using an interface
board and communications portal enabling synchronous communication protocols. A
signal conditioning box allows connections to the ECC's analog signals that are
conditioned with custom electronic components. Minor but necessary components
include pressure and temperature sensors, and valves and solenoids to direct the flow of
laboratory grade air. Calibration validity is accomplished by comparing the measured
ECC ozone partial pressure against a reference partial pressure obtained with the UV
photometer.


Fig. 2, from an unpublished technical note (Baldwin, private communication), illustrate
the steps necessary to achieve a consistent calibration. By following the sequential flow
diagram shown in Fig. 2, upper panel, the operator can better understand the sequence of
tests. Each shape in the diagram is associated with a graphical window displayed on the
monitor, as are notices that pop-up to instruct or direct the operator. The computer
controlled digital bench follows the ECC preparation procedure in place at NASA
Wallops Island at the time of the system's fabrication. Each ECC is recognized by its
manufacturing date and serial number and includes the manufacturers test data. Changes
to the steps are possible anytime through software reprogramming. Operationally, the
preparation sequence begins by verifying whether ECC cells are new or were previously
conditioned. A different path is followed for either condition. New cells are flushed with
high ozone prior to manually adding KI solution. Cells previously having had solution
added skip over the high ozone step to determine the first background current. Following
the first background check the remaining steps are completed. Other measurements



accumulated with the digital bench include motor voltage, motor current, pump
temperature, and linear calibration at seven levels (0-300 nb).  Program steps are
displayed on the computer monitor with real-time information. All data are archived and
backup files maintained.

Fig. 2, lower panel, illustrates the functional diagram detailing the essential operation of
the digital calibration bench. Software control is shown in blue and air flow in green.
Laboratory zero-grade dry air or desiccated compressed air is introduced into the ozone
generator (TEI Generator) where a controlled amount of ozone is produced. The ozone
flows simultaneously to the ECC cells and to the ozone analyzer (TEI Analyzer). The
analyzer provides the reference partial pressure.

The measurement of the air flow and the corresponding time permits a precise flow rate
to be determined. In contrast, the manual method uses a stop watch to estimate when 100
ml of air has flowed into a chamber. An experienced operator, using a volumetric bubble
flow meter should be able to measure the time to within 1 second, possibly better.
Although great care is exercised to obtain this measurement an error of one second is
equivalent to an approximately three percent error in the measurement of ozone partial
pressure. Further, the manual method requires that the effect of moisture from the bubble
flow meter's soap solution be accounted for; flow rates determined with the digital
calibration bench do not require a correction for moisture. Unfortunately, the calibration
bench cannot determine the pump efficiency correction (PEC); this is taken into account
differently. For a number of years, the ECC's PEC was physically measured at Wallops
Island using a specially adapted pressure chamber (Torres, 1981). This system no longer
is available. However, from its many years of use an extensive number of measurements
are available. A sample of 200 pressure chamber measurements were averaged to obtain a
unique PEC that was adopted for use at Wallops Island.

After eliminating deficiencies and improving functionality the automated system was
tested while obtaining research data, primarily comparisons between different KI solution
concentrations. Unfortunately, comparison with manually prepared ECC's was never



contemplated. Calibration from 0 nb to 300 nb generally exceeds the nominal range of
atmospheric ozone partial pressure. Calibration steps are made in 50 nb increments but
larger or smaller increments are possible with minimal software reprogramming.
Differences between ECC and reference measurements, if seriously large, provide an
alarm to possibly reject the ECC, or after further study the differences between the ECC
and reference calibration might be considered as a possible adjustment factor that would
be applied to observational data.

2.2  Operational Procedure

ECC preparation procedures at Wallops Island are carried out five to seven days prior to
preparing the ECC for flight. The pump, anode and cathode cells, and Teflon tubing are
flushed with high amounts of ozone to passivate their surfaces and is followed by
flushing with zero-grade dry air followed by filling of the cells. The cells are stored until
ready to be used.

Operation of the automated system is simple, requiring only a few actions by the operator
that include obtaining the first background current, air flow, 5 µA or high ozone (170 nb)
test, response test, second background current, linear calibration between 0 nb and 300
nb, and the final background current. Two cells can be conditioned nearly
simultaneously. i.e., the program alternates measurements between ECC's.

The operator must first determine whether the cell being conditioned had already been
filled with KI or never was filled. Whatever the status of the cell (wet or dry) the operator
must enter the identification information before proceeding. When a new, or a dry cell is to
be processed the digital calibration bench initiates high ozone flushing. The program alerts the
operator to turn on the high ozone lamp after which V3 of Fig. 2, lower panel, is switched to high
ozone. The unit checks that ozone is flowing and after 30 minutes the program switches to zero
air for 10 minutes and V3 switches back to the ozone generator. When completed, the operator is
prompted by an instructional message on the monitor screen to fill the anode and cathode cells
with the proper concentrations of potassium iodide (KI) solution. The cells are stored until ready
for further conditioning and calibration before being used to make an observation. Considering
that the ECC cell had been filled earlier with solution the digital bench instruction by-
passes the high ozone flushing. Ozonesonde identification is entered, as above. The
operator, after fresh KI has been added to the cell, is prompted on the monitor screen to
begin the first background current measurement. In either case, whether a dry cell for
which flushing is complete, or a wet cell ready for calibration, the procedure starts with
clicking the OK button displayed on the monitor screen. After 10 minutes of dry air the
background current is recorded. The background current record contains the following
information: date, time in 1-2 second intervals, motor current, supplied voltage, pump
temperature, and cell current. As the measurement is being made identical information is
displayed graphically on the monitor. Following the background test all further steps are
automatic.

Continuing to follow the steps outlined in Fig. 2, upper panel, the measurement of the air flow is
accomplished on one ECC pump at a time by switching V1, shown in Fig. 2, lower panel, to the
mass flow meter and at the same time V2  is switched to the glass manifold (ozone generator).
When completed, V1 is switched back to the glass manifold and V2 is switched to the flow meter
and the flow rate of the second cell is carried out. The air flow is output in sec/100 ml. The
information stored includes: date, time in seconds at intervals of 7-8 seconds, mass flow meter
temperature, atmospheric pressure, flow rate, and supply voltage.

Response of the ECC to ozone decay requires setting the ozone generator to produce 170 nb
ozone partial pressure (approximately 5 uA). As ozone is produced the ozone level increases until
the set level is reached. The elapsed time to reach this level is noted. The 170 nb of ozone is the
reference level used to initiate the response test. After recording 170 nb of ozone for one minute
the ECC response check begins. To measure the response, the cells would have to be switched to
zero air quicker than the cell responds. This is accomplished by switching both cells (assuming
two cells are being calibrated) to the mass flow meter, the source of zero air. This is more
efficient than setting the generator to zero and waiting for the manifold and residual ozone in the
system to reach the zero level. Thus, VI and V2 of Fig. 2, lower panel, are switched to the mass
flow meter for immediate zero air and the program triggers a timer. The decreasing ozone is
measured and recorded at five points used to reflect the cell response. As the ozone decays,
measurements at 3-4 second intervals provide a detailed record of the response while also being
displayed real-time on the monitor. The detailed record is hacked by the program at five points (1,



2, 3, 5 and 10 minutes) successively and calculates the percentage of ozone change that occurred
at the one-minute point which should be 80-90 percent lower than the reference of 170 nb. V1
and V2 are switched back to the ozone generator and the next 10-min background current
measurement begins. The response record contains the following: date, time in seconds, motor
current, supply voltage, temperature, mass flow, cell current, and atmospheric pressure. Data are
displayed on the monitor in real-time.

The ECC cells have been conditioned and are ready for the linear calibration. The 0 nb to 300 nb
calibration is performed. Step changes begin with 0 nb, followed by measurements at 50, 100,
150, 200, 250, and 300 nb. Each step requires approximately 2-3 minutes to complete allowing
time for the cell to respond to each ozone step change. The linear calibration includes the
reference measurement made simultaneously with the ECC measurement. After the upward
calibration reaches the 300-nb level the calibration continues downward, to 0 nb. The
measurements are displayed on the monitor for the operators use and also sent to an Excel file.
Generally, the downward calibration experiences small differences from the upward calibration
Only the upward calibrations are used.

Following the linear calibration, the final background current is obtained. As before this requires
10 minutes of zero grade dry air before making the measurement. The data are recorded.

A summary is provided of the calibration giving supply voltage, motor current, flow rate, pump
temperature, response, and three background currents.

3  Digital Calibration Bench Practical Application

Repetitive comparison operations can be carried out with the digital calibration bench as
often as necessary. This could result in a potential cost saving as there would not be a
need to expend radiosondes, ECC's, and balloons. The testing with the digital calibration
bench is limited to sea level conditions

3.1  Digital Calibration Bench (General)

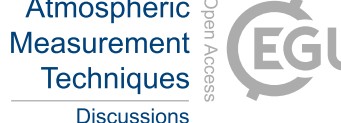

Quasi-simultaneous testing of two ECC's is possible, enabling comparisons of different
concentrations of KI solutions. Comparison of 2.0-, 1.5-, 1.0-, and 0.5- percent KI
concentrations demonstrated that agreement with the reference improved with lower
concentrations. Only the SPC 6A ECC's with 1.0 percent KI solution and full buffer
(1.0%,1.0B) and 0.5 percent KI solution and one-half buffer (0.5%,0.5B) concentrations
are discussed, however.

Testing indicated the pressure and vacuum measurements were nominal, some
insignificant variation occurred but was not a cause of concern. Pump temperatures,
controlled by the room air temperature, varied 0.1ºC to 0.2ºC. Motor currents showed
some variation, some measured over 100 µA, suggesting a tight fit between the piston
and cylinder. For example, one ECC motor current initially was 100 µA, a second
measurement a week later the reading was 110 µA, a final reading after running the
motor for a short time was 96.5 µA. Flow rates fell within the range of 27 to 31 seconds
per 100 ml, a range comparable to flow rates manually measured with a bubble flow
meter. Background currents were consistent. The lowest background current allowed by
the digital bench is 0.0044 µA. The final background currents often were somewhat
higher than background currents experienced with manual preparation, generally 0.04 µA
on average. Final background currents obtained prior to a balloon release was in the
range between 0.01 and 0.02 µA. Finally, the response of all the cells was good, falling
within the necessary 80 percent decrease within less than one minute. Graphically
checking a small sample of high-resolution responses found some variation as ozone
decreased to 0 nb. The linear calibration (0-300 nb), is useful for comparing different KI
concentrations.

3.2  Calibration and Potassium Iodide (KI) Solution Comparisons

As a practical example of the usefulness of the digital calibration bench is its capability to
nearly simultaneously obtain measurements from two ECC's, one prepared with
(1.0%,1.0B) and the second with (0.5%,0.5B). Conditioning of the ECC's followed the
steps given in Fig. 2, upper and lower panels. In the free atmosphere ozone partial



pressures usually range up to 150 nb to 200 nb. Linear calibrations to 300 nb are
obtained, although a lower range may be reprogramed.

Figure 3 is a graphical example of differences between the reference ozone and the
measurements of (1.0%,1.0B) and (0.5%,0.5B) KI concentrations. Rather than showing
the differences from a single measurement, a sample of 18 digital bench measurements
were averaged to give a more representative set of differences. Fig. 3 suggests that the
two concentrations measured nearly identical amounts of ozone between 0 nb and 80 nb.
Both curves begin to separate and diverge above 80 nb. The averaged data at 100 nb
indicate that (1.0%,1.0B) is 3.6 nb, or 3.6 percent higher than the reference and
(0.5%,0.5B) is 0.4 nb, or 0.4 percent higher; at 150 nb the difference is 6.7 nb, or 4.3
percent and 1.7 nb or 1.1 percent higher, respectively; at 200 nb the difference for
(1.0%,1.0B) is 11.1 nb, or 5.5 percent and (0.5%,0.5B) is 4.8 nb or 2.4 percent higher,
respectively. A check at the 300 nb level indicated (1.0%,1.0B) was 7.2 percent above the
reference and (0.5%,0.5B) was 3.7 percent above. The ECC with (0.5%,0.5B) KI
concentration is closer to the reference than (1.0%,1.0B) KI . Both ECCs' partial pressure
curves have a slope greater than 1 trending toward higher amounts of ozone when
compared to the reference value as ozone partial pressure increases. It is clear from the
digital bench testing that the (1.0%,1.0B) KI solution increases at a faster rate than the
(0.5%.0.5B) solution as ozone partial pressure increases. This non-linearity is not
explained here. The intent of the examples is merely illustrative of the advantage
provided by the digital bench to examine ECC behavior. Further, Fig. 3 indicates that the
1.0 percent KI measurement is further from the reference than the 0.5 percent KI while
the percentage difference between the two concentrations is nearly constant at 3.2
percent, or in terms of a ratio between the two solutions, 0.968. Referring to the SPC
ozonesondes compared during BESOS, Deshler et al (2017, Fig.5 and Table 2) indicates
non-linearity between the (0.5%,0.5B) and (1.0%,1.0B) KI solutions and similar ratio
values, 0.970/0.960 .

The digital calibration bench turned out to be an ideal tool to obtain repeated ECC
calibrations. The digital bench can calibrate two ECC's nearly simultaneously reducing





the need to expend costly dual-ECC balloons. A negative aspect, possibly, is calibration
occurs under sea level conditions so cannot provide knowledge of ECC behavior under
atmospheric conditions. A series of calibrations were performed over a period of three
weeks. Two new ECC's were prepared with (1.0%,1.0B) and (0.5%,0.5B) KI solutions.
Although a number of time-separated calibrations were conducted, only one three-week
test is shown in Fig. 4a, b, c. The result shown is characteristic of similar calibrations
performed over a similar number of weeks. The cells were flushed and fresh KI solutions
were used with each weekly test. Calibration over the full range, 0-300 nb was carried
out, only the 300 nb partial pressures are discussed. During the first week, Fig. 4a, the
(1.0%,1.0B) KI solution was approximately 21 nb, or 7 percent higher than the
corresponding reference value. The (0.5%,0.5B) KI solution was about 6-7 nb or about 2
percent lower than the reference value. A second calibration one week later, designated
week two in Figure 4b, showed the ECC with the (1.0%,1.0B) KI solution had moved
further away from the reference, about 27-28 nb or 9 percent higher (approximately 6-7
nb higher than during week one), while the ECC with the (0.5%,0.5B) KI was now 12 nb
or 4 percent higher than the reference. A third calibration, week three in Fig. 4c, showed
both ECC calibrations had moved again. The (1.0%,1.0B) KI calibration increased an
additional 2 nb and was now about 30 nb, or 10 percent higher than the reference. The
ECC with (0.5%,0.5B) KI increased an additional 1 nb and now was 13 nb, 4 percent
higher than the reference value. Providing an explanation for the changes observed
between week one and week three is difficult. Changes that might be due to improper
preparation and conditioning procedures is not considered since, by definition, the digital
bench is consistent in how ECC's are prepared, i.e., it is expected that carrying out the
preparation would be repeatable from week-to-week. Consideration also must be given to
the fact that the ECC has a memory. It is very possible that calibrations taking place
following week one could still be under the influence of the previous measurement due to
some impurity residuals present on the ion bridge. On the other hand, the changes could
simply be a normal evolution of typical ECC performance.

The curves shown in Fig. 4a, b, and c merely show the calibrated ECC offset relative to a
reference, or "true" partial pressure. To bring the ECC measurements into



correspondence with the reference suggests that downward adjustment should be applied
to each curve. However, how should such time-separated calibrations be treated; should
only the final calibration (e.g., week 3) be used or an average of the three calibrations.
Regardless, after obtaining a large sample of similar digital bench measurements it would
be possible to design a table of adjustments relative to ozone partial pressure to be used to
adjust in-flight ozonesonde measurements. However, the calibrations are made at sea
level and cannot account for the influence of atmospheric pressure and temperature.
Nevertheless, any adjustment seemingly would be in the right direction and would aid in
obtaining more representative ozone values.

Although digital bench calibration comparisons are instructive, important comparisons
have been made between ECC's and reference instruments using other methods. ECC
measurement comparability have been quantified through in situ dual instrument
comparisons (Kerr et al, 1995; Stubi et al, 2008; Witte et al, 2019), laboratory tests at the
World Ozone Calibration facility at Jülich, Germany (Smit et al, 2004, 2007, 2014) and
by occasional large balloon tests such as BOIC (Hilsenrath et al, 1986), STOIC (Kohmyr
et al, 1995) and BESOS (Deshler et al, 2008). BESOS provided important performance
information about the SPC 6A ECC and the EnSci ozonesondes. Only the SPC 6A ECC
is discussed. However, these complicated large balloon experiments that seem to occur
every 10 years are expensive. The environmental chamber used in the Jülich tests covers
a full pressure range but is also expensive to use. The purpose here is to show a
calibration method that is simpler to use and provides calibration that includes a useful
reference value, and is complementary to other methods, such as employed in the Jülich
Ozone Sonde Intercomparison Experiment (Smit et al, 2007).

BESOS was conducted from Laramie, Wyoming during April 2004, employed a large
balloon carrying a gondola fitted with 12 dedicated ozonesondes. The gondola also
carried an independent power supply, a multiplexer/transmitter, and a UV photometer.
The photometer (Proffitt and McLaughlin, 1983) was used for over 20 years in various
tests conducted at the Jülich facility. Other instruments included on the gondola are not
germane to the present discussion. The ECC's were divided into two groups, each group

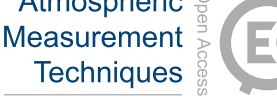

consisting of six SPC-6A and six EnSci ECC's. Each group of six ECC's was further
partitioned into two sub-groups. One sub-group was prepared with 1.0 percent fully
buffered KI solution, the second sub-group was prepared with 0.5 percent KI and one-
half the buffer. Only the two SDC-6A sub-groups and the UV photometer are of interest
to this discussion. The BESOS test design allowed comparison of: the differences
between (1.0%,1.0B) and (0.5%,0.5B) KI solutions; the differences between SPC-6A and
EnSci ECC's; and, the differences between both ECC types and the reference photometer
(Deshler et al, 2008).

The photometer data were noisy during the early portion of the flight and did not provide
reliable data. The remainder of the flight experienced intermittent data loss, but overall
sufficient data were available to carry out an analysis, particularly in the stratosphere
(Deshler et al, 2008). Partial pressures lower than 60 nb are not discussed. The data were
separated into two displays of ozone partial pressures as shown in Fig. 5a and Fig. 5b.
The filled diamonds, filled triangles, and filled circles illustrate the ECC/photometer
relationship.

The least-squares method was used to fit the ozonesonde data in Fig. 5a, b. The ECCs'
with the 1.0 percent KI, shown in Fig. 5a, measured increasingly more ozone than the
reference as the ozone partial pressure increases. There is 3 percent more ozone measured
at 100 nb, and 5 percent more ozone measured at 150 nb, than the photometer reference.
This is within reasonable agreement with the digital calibration bench estimates, of 3.6
and 4.3 percent, respectively. The relationship between SPC-6A ECCs' prepared with 0.5
percent KI solution and the UV photometer, shown in Fig. 5b, is in closer agreement with
the UV photometer than the 1.0 percent KI solution. The 0.5 percent partial pressures are
mostly the same as the photometer values, but a small negative slope can be discerned.

In the 1998-2002 period the Wallops ozone station released a number of dual-ECC
balloons, twelve pair successfully provided measurements to 30 km, and higher. The
ECC's were attached about 35 meters below the balloon and each ECC was separated 2
meters. Each pair was composed of an ECC with (1.0%,1.0B) and (0.5%,0.5B) KI



solutions. The profiles were averaged, and are displayed in Fig. 6. The profiles are
interesting in that the 1 percent ECC and the 0.5 percent ECC measured virtually the
same ozone partial pressure until reaching 70-80 nb, at an atmospheric pressure of
approximately 65 hPa. At this level the (0.5%,0.5B) ECC began to measure less ozone
than the (1.0%,1.0B) ECC. A similar feature was noted in Fig. 3 where the separation of
the ECC's with different concentrations occur at about 80-90 nb. Fig. 6 shows the
maximum ozone level was about 140 nb, near 22 hPa, where (0.5%,0.5B) KI measured
approximately 10 nb, or 7 percent less ozone than that of the (1.0%,1.0B) KI
concentration. This difference is approximately 4 percent higher than the result given by
the digital calibration bench results of Fig.3, where, at 150 nb, the difference between the
ECC 1 percent KI and ECC 0.5 percent is 3.2 percent.

Given that the digital bench tests revealed the (0.5%,0.5B) KI solution is in closer
agreement with the reference measurement than the (1.0%,1.0B) solution suggested that a
KI solution with a weaker concentration may possibly give even closer agreement. A
small number of dual ECC tests were carried out. The decision was made to try a solution
of 0.3 percent with one-third buffer (03%,0.3B).  Six sets of ECC's were prepared for
calibration. Each dual ECC test consisted of one ECC prepared with (1.0%,1.0B) KI
solution  and one with (0.3%,0.3B) KI solution. The digital bench comparison result
disclosed the (1.0%,1.0B) result replicated the earlier results discussed above. As
assumed, the lower concentration was nearly equal to, or slightly less than the reference.
Average values derived from the six tests are shown in Fig. 7. To corroborate the bench
results three balloon-borne dual ECC sondes were flown, each with 1.0 and 0.3 percent
KI solutions. Unhappily, the results were inconclusive: one flight showed (0.3%,0.3B) to
be higher than (1.0%,1.0B), a second flight showed it to be lower, and the third flight
showed (0.3%,0.3B) to be nearly the same value. Although the 0.3 percent solution might
appear to be a better choice additional tests are necessary.

4  Summary



The concept of an automated method with which to pre-flight condition and calibrate
ECC ozonesondes was originally considered by MeteoSwiss scientists over 20 years ago.
Drawing on their expertise, a facility designated as the digital calibration bench was
fabricated at NASA Wallops Flight Facility between 2005-2007. The digital  bench was
put to use immediately to study ECC performance, conduct comparisons of different KI
concentrations, enabled ECC repeatability evaluation, as well as calibrating the ECC over
a range of partial pressures, including associated reference values. Tests conducted with
the digital bench were performed under identical environmental conditions. The digital
bench eliminates the expense and time associated with making similar tests in the
atmosphere.

Early use of the digital bench was to calibrate ECC's, prepared with (1.0%,1.0B) KI
solution, over a range of partial pressures from 0 nb to 300 nb. Comparison between
ECC's with (0.5%,0.5B) and (1.0%,1.0B) KI solution and comparing their measurements
with simultaneously obtained reference values revealed both KI solution strengths were
measuring more ozone than the reference. There was an increasing difference between
the ECC's and the reference as the partial pressure increased. For example, the ECC
measurements slope upward to increasingly larger differences from the reference ozone
measurements, i.e., increasing from 4.3 percent higher partial pressure at 150 nb (Fig. 3)
to about 7 percent higher at 300 nb.

An instruments ability to repeat the same measurement is important, however,
ozonesondes are used only one time. (There are exceptions when an occasional
instrument is found and returned, but, unfortunately because of Wallops Island's coastal
location nearly all sonde instruments fall into the Atlantic Ocean rendering them unfit to
be reclaimed). The digital bench provided the opportunity to obtain repeatable
calibrations of the ECC. Results from testing ECC cells over a period of three weeks, one
test each week, showed the calibration changed, e.g., about 10 percent for 1.0 percent KI
and 4-5 percent for the 0.5 percent solution.





Results from the digital bench also corroborate differences found between SPC 6A
ECC'c flown on BESOS and also with dual-instrument flights flown at Wallops Island.
The difference between ozonesondes at a pressure of 22 hPa showed the (0.5%,0.5B)
ECC to be about 10 nb lower than the (1.0%,1.0B) ECC.

The digital calibration bench provides a capability to apply a variety of test functions
whereby the valuable information gathered helps to better understand the ECC
instrument. Evaluating SPC ECC performance using an automated method diminishes the
requirement for expensive comparison flights. The tests performed, i.e., KI solution
differences, calibrations over a time period, and dual-instrumented balloon flights, were
consistent, giving similar results. The tests described in this paper are simply examples of
the digital bench utility. Furthermore, not mentioned earlier, the digital calibration bench
preparation facility potentially could contribute to an understanding of separating ECC
variability from atmospheric variability. Thus, the automated conditioning and calibration
system provides valuable information, and as a useful tool should continue to be a
valuable aid.

5  Data Availability
Data are available from the authors.

6  Author Contribution
The first author acquired and prepared the data for processing and the second author was
instrumental in certifying the digital calibration bench was working properly. Both
contributed equally to manuscript preparation.

7  Competing Interests

The authors declare they have no conflict of interest.

8  Disclaimer



None

9 Acknowledgments
We acknowledge the successful use of the digital calibration bench to the skillful efforts
of Gilbert Levrat (retired) of the MeteoSwiss site Payerne, Switzerland for his foresight
in designing the original bench and its simplicity, and to Tony Baldwin (retired) of
NASA Wallops Flight Facility for his electronic skill and programming expertise.

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




11 Figures

Fig01. Digital calibration bench showing operational configuration and mounting
position of two ECC ozonesondes. The major instrumentation includes ozone generator
and analyzer, computer, flow meter, and glass manifold.

Fig02. Digital calibration bench diagrams showing a) sequential steps, and b) functional
steps.

Fig03. Simultaneous measurements of ECC ozonesondes, prepared with different KI
solution concentrations. Average differences are shown between 1.0 and 0.5 percent KI
strengths. The blue curve represents (1.0%,1.0B) KI, the red curve (0.5%,0.5B) KI and
the reference curve is shown in black. Calibrations are made in 50 nb steps from 0 nb to
300 nb.

Fig04. Calibrations of two ECC ozonesondes, one using 1.0 percent KI solution and the
other 0.5 percent KI,  over a three week period.

Fig05. Correlation between SPC 6A ECC ozonesondes and UV photometer
measurements obtained during the BESOS mission: a) 1.0 percent KI solution, and b) 0.5
percent KI solution.

Fig06. Average ozone profiles from 12 pair of SPC 6a ECC ozonesondes indicating, at
the 22 hPa pressure level, that the (0.5%,0.5B) ECCs' measured 7-8 nb less ozone,
approximately 5 percent less, than the (1.0%,1.0B) ECCs'.

Fig07.  Digital calibration bench results between (1.0%,1.0B) solution, blue curve, and
(0.5%,0.5B) solution, red curve; the reference curve is shown in black.





Fig 01.


# DIGITAL CALIBRATION BENCH

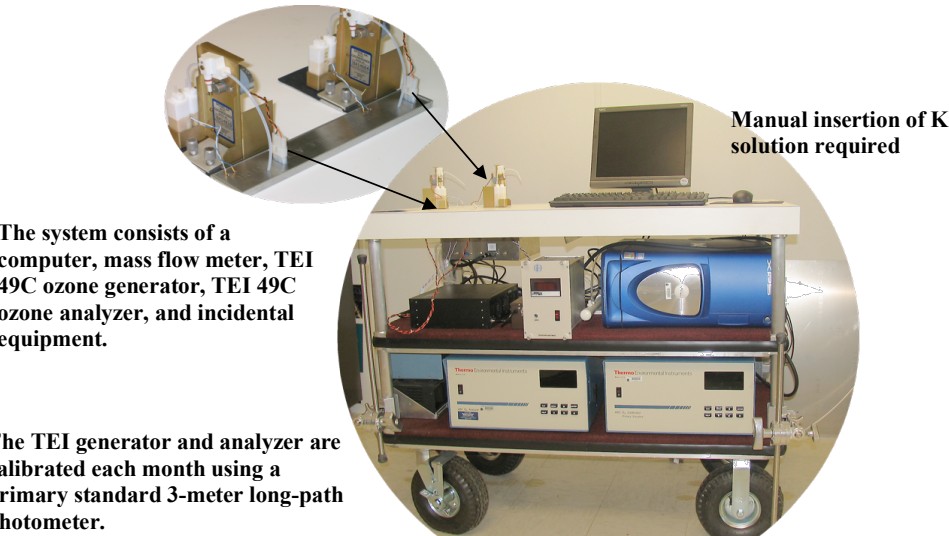

**Manual insertion of KI solution required**

**The system consists of a computer, mass flow meter, TEI 49C ozone generator, TEI 49C ozone analyzer, and incidental equipment.**

**The TEI generator and analyzer are calibrated each month using a primary standard 3-meter long-path photometer.**



Fig 02.

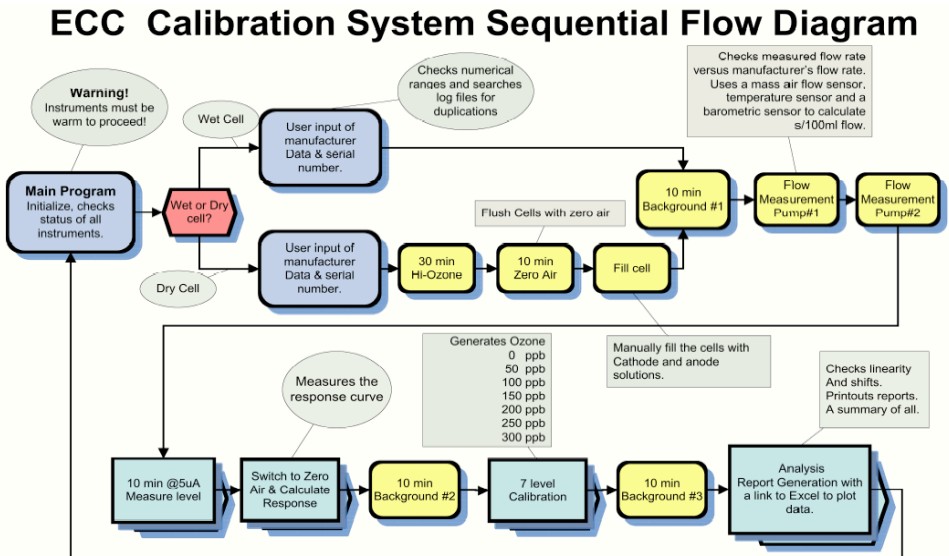


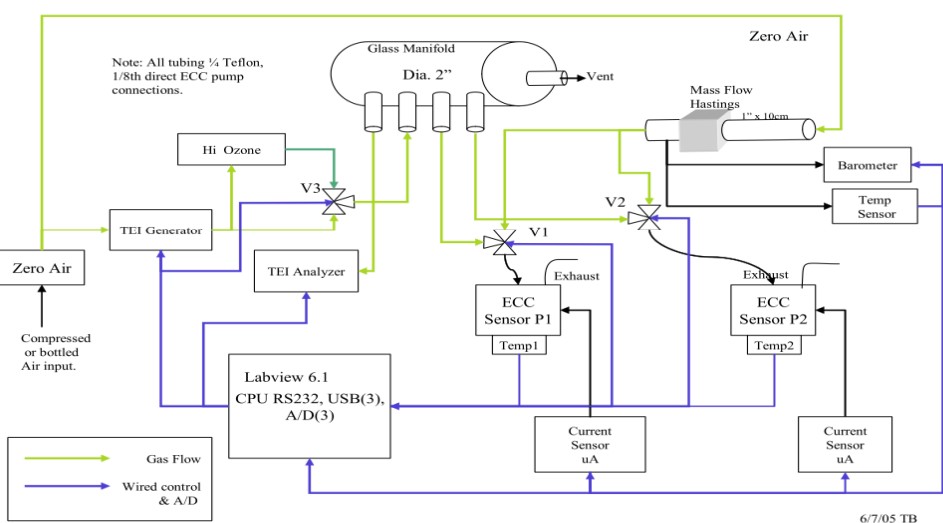


Fig 03.

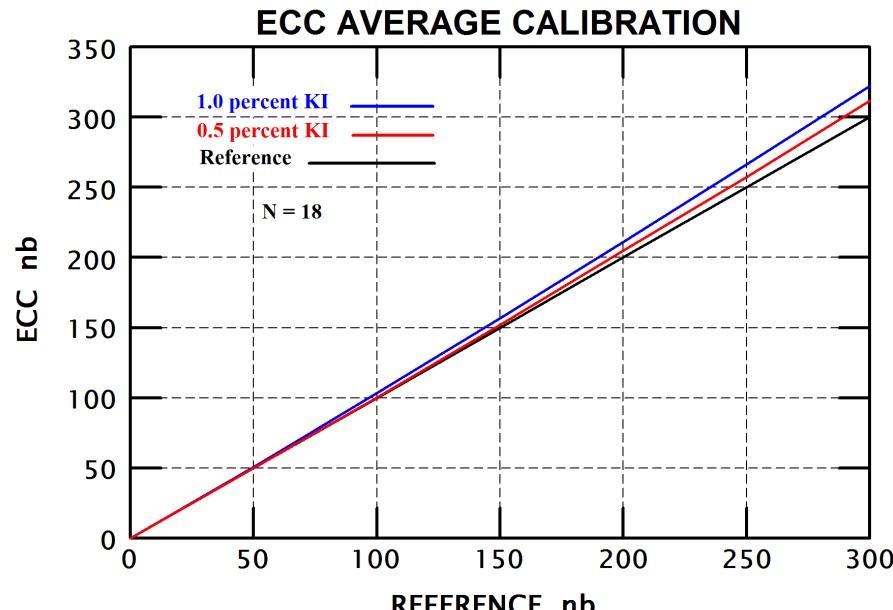




Fig 04.

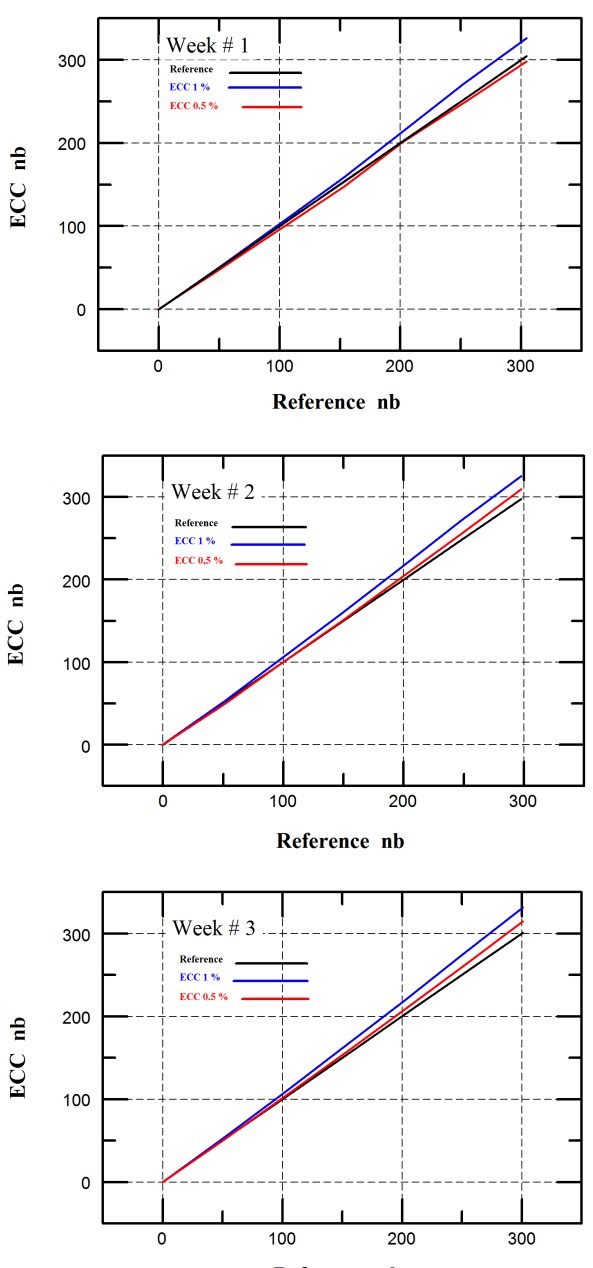






Fig 05.



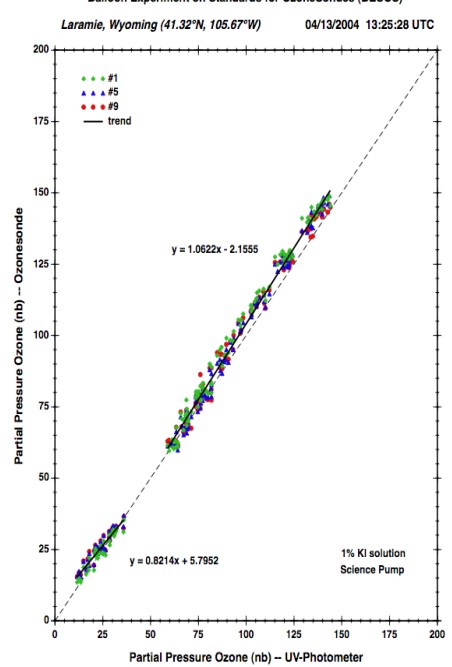
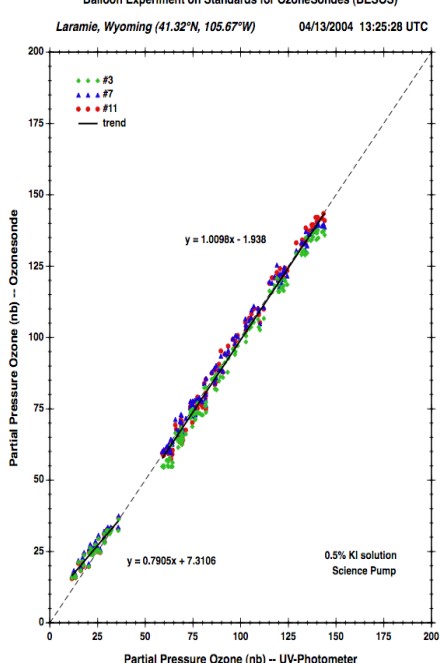



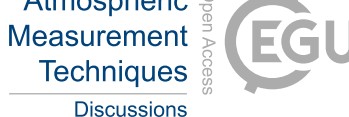

Fig 06.

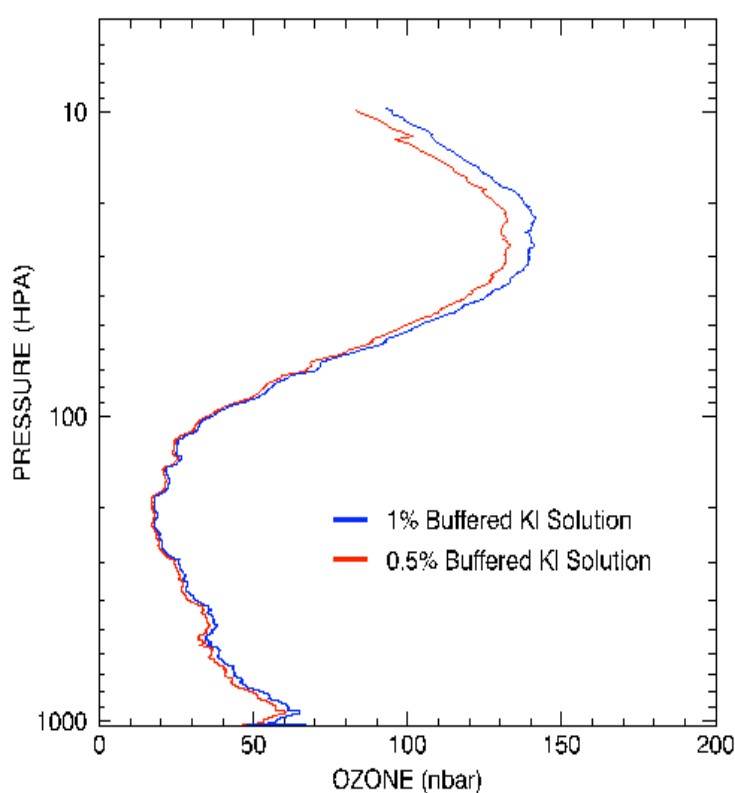






Fig 07.

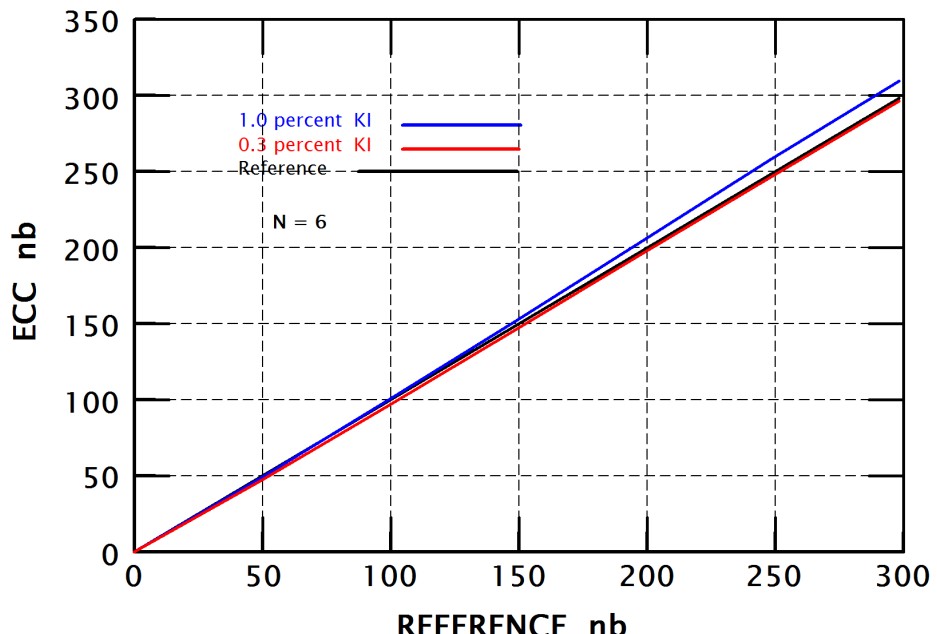
