# Peer review of "An Automated Method for Preparing and Calibrating"

_Atmospheric Measurement Techniques, 2019_

## Referee Comment (RC1) · Anonymous Referee #1 · 12 Oct 2019

General comments

The manuscript can be divided into two parts. In the first part the authors describe the design of a digital calibration bench for ECC ozonesondes in use since 2007 at the NASA/GSFC/Wallops Flight Facility. In the second part the digital calibration bench is used to test Science Pump Corp. 6A ECC ozonesondes with two different sensing solution types. In the first part the digital calibration bench itself is good described. Preparation of ozonesondes using such a device is superior to a manual preparation in particular when a UV photometer as a reference is used. This description alone qualifies for a publication in AMT. With respect to the second part it is not clear to me

whether this part is only written to demonstrate the prospects of the bench as indicated e.g. at line 326 or to make valuable scientific statements. In a demonstration mode large portions, e.g. the BESOS discussion, can be omitted. For scientific statements the whole second part offers some potential for improvements, i.e. a better statistic and an error analysis. However, in total I recommend the publication of the manuscript after some minor revisions.

Specific comments

1. The title of the manuscript is dealing with the first part only. The title should address both parts in case the second part is not for demonstrations only.

2. As pointed out several times the aim of the digital calibration bench was to investigate the behavior of ECC ozonesondes and to compare different configurations in a consistent and resources conserving manner replacing e.g. dual soundings. Although the advantage of reducing subjectivity compared to the manual preparation is mentioned, a clear statement is missing, that the bench is used at the Wallops Flight Facility for routine soundings (since when?), too. In this frame, one can address the fact that such calibration benches would be of benefit in particular for the ozonesonde records at remote sites with frequent exchange of operators (neglecting the needed financial effort).

3. Line 108: What means "similar" to the MeteoSwiss version? Are there improvements?

4. Line 159: Please list manufacturer, sensor type, measurement principle of the flow rate measurement device. The same is desired for the UV photometer.

5. Line 207: I am sure that the authors do know that the cathode and anode cells have to be filled in the right sequence and that the instructions are accordingly. Please give a small hint.

6. Line 233: "After recording 170 nb of ozone for one minute". Fig. 2 (upper panel)

tells "10 minutes" instead. I assume the 10 minutes are true.

7. Line 271: I suggest: ". . . bench is limited to pressure and temperature ranges appearing at sea level."

8. Lines 282-298: In order to classify some statements in this paragraph the statistical background, i.e. the number of investigated sondes, is needed already here. E.g. the background current can be batch dependent, which should relativize the statement at lines 291-293.

9. Lines 335: I would agree to substitute "ideal" by "good", since a negative aspect is mentioned right after.

10. As already mentioned before, the second part suffers from a missing statistical error analysis. Presented are only averaged data without error bars (or single cases). Without knowing the statistical errors it is impossible to justify whether the number of underlying cases is sufficient large.

11. Lines 341-342: Why is only one example shown here? For all other cases the averages were shown.

12. Lines 369-370: A first answer would be the final calibration. However, again, it would be helpful to see the other examples.

13. Lines 393-424: Is this (incl. Fig. 5) a new analysis not conducted in the BESOS publication before? BESOS outcomes had been already discussed at lines 330-333. However, a comparison to JOSIE2000 is missing. Why?

14. Lines 430-433: I disagree with the statement ". . . measured virtually the same ozone partial pressure until reaching 70-80 nb . . .". Obviously, the 0.5% sondes measure significant less ozone in the lower troposphere, too. A plot showing the differences in relative units would be interesting.

15. A last comment for the future use: The test environment is bound to the surface

conditions. One might learn more how to use the bench calibrations within these limits by combining them with subsequent dual flights or chamber experiments like JOSIE.

Technical corrections

1. Line 45: Please use SI units throughout the manuscript, i.e. mPa instead of nb for the ozone partial pressure.

2. Line 49: Write out the acronym BESOS in the abstract, too.

3. Line 88: Delete one "the".

4. Line 250-252: The steps are in ozone partial pressure. In Fig. 2. upper panel the steps are given in mixing ratios. What is actually used?

5. Fig. 2 lower panel: - The blocks with "TEI Generator" and "Hi Ozone" seems to be misleading. As far as I understood the ozone is generated inside the generator and not outside. I guess the TEI Generator has one outlet, which sends Zero Air, when the generator inside is off, and Hi Ozone, when the generator is on. In that case V3 would be needless (or somehow hidden in the generator). Or, the generator has two outlets, one for Zero Air and one for Hi Ozone. In that case V3 makes sense. What is true? - If you use a different color for Hi Ozone please explain it in the legend. - The blue arrows at the barometer and the two current sensors indicate that the computer is triggering these devices. Is that right? - The writing of the word "Exhaust" near ECC Sensor P2 should be shifted to the right to the real exhaust. - How does the information of the mass flow measurement go into the computer? Is there a wired control connection (please indicate it in the diagram) or is it manually transferred by the operator (please note it in the main text)?

6. Fig. 3: Why does the plot differ somewhat from the first submitted version? Please comment in your reply only and not in the manuscript.

7. Fig. 6: Please add "N = 12" in the plot to be consistent with the other plots.

---

## Referee Comment (RC2) · Anonymous Referee #2 · 22 Oct 2019

This is a worthwhile paper, and should be published. I have a number of minor concerns that the authors may wish to address first, however.

Pg. 4, lines 92-101: Some mention of the efforts of the O3S-DQA initiative (Smit et al., 2012; Smit and ASOPOS panel, 2014) would be appropriate here. Perhaps even some of the recent re-evaluation papers (Tarasick et al., 2016; Van Malderen et al., 2016; Witte et al., 2018; 2019; Sterling et al., 2018) would not be out of place. The references Barnes (1982) and Barnes et al (1985) for sonde accuracy are rather old, and there are better ones, which the authors know as they co-authored some of them. There is a good summary in the forthcoming ASOPOS-2 report, also published as a

paper in review for Earth and Space Science (Tarasick et al., 2019).

Pg. 4, line 97: "whether measured". Might insert "it is" to make comprehension easier for non-native speakers.

Pg. 4, line 98: "the use of the appropriate potassium iodide (KI) concentration". While the KI concentration does have an effect, the uncertainty really lies with the stoichiometry of the KI reaction with ozone, as well as unwanted side reactions with the phosphate buffer. Losses of ozone and/or iodine in various ways should be included in this list, and motor speed might also be so included, since motors have changed in recent years.

Pg. 6, lines 159-167: What is the uncertainty of the automated flow rate measurement? This discussion seems to treat it as zero! The volumetric bubble flow method is quite accurate (and as a method traceable to physical constants, is typically used to calibrate automatic devices). Operator uncertainty is about 0.1-0.3% (Tarasick et al., 2016), less than 1/10 of what the authors suggest; the automated Gilibrator is only slightly better (if used properly).

Pg. 8, line 230: Insert "Measuring the..." before "Response". Line 242: "hacked" is slang; moreover it's not clear what is meant.

Pg. 9, line 271: Text missing here?

Pg. 10, lines 276-278: Should cite Johnson et al. (2002) here.

Pg. 11, lines 325-326: On the other hand, it's explained in great detail in Johnson et al. (2002). Why not refer to that?

Pg. 13, lines 369-370: Good question. The variation shown suggests a variability of about 5%, at least for the 0.5% solution. That is rather large, and serious investigation of it might add a lot to current understanding of ECC uncertainties, since, as the authors point out, such investigations are much easier to do than experiments at the World Ozone Calibration facility at Jülich.

References

Smit, H.G.J., and ASOPOS panel (2014), Quality assurance and quality control for ozonesonde measurements in GAW, WMO Global Atmosphere Watch report series, No. 121, 100 pp., World Meteorological Organization, GAW Report No. 201 (2014), 100 pp., Geneva. [Available online at https://library.wmo.int/pmb_ged/gaw_201_en.pdf]

Smit, H.G.J., S. Oltmans, T. Deshler, D. Tarasick, B. Johnson, F. Schmidlin, R. Stuebi and J. Davies (2012), SI2N/O3S-DQA activity: Guidelines for homogenization of ozone sonde data, Activity as part of SPARC-IGACO-IOC Assessment (SI2N) "Past Changes In The Vertical Distribution Of Ozone Assessment", 2012. available at: http://www943das.uwyo.edu/%7Edeshler/NDACC_O3Sondes/O3s_DQA/O3S-DQA944Guidelines%20Homogenization-V2-19November2012.pdf

Sterling, C. W., B. J. Johnson, S. J. Oltmans, H. G. J. Smit, A. F. Jordan, P. D. Cullis, E. G. Hall, A. M. Thompson, and J. C. Witte (2018), Homogenizing and estimating the uncertainty in NOAA's long -term vertical ozone profile records measured with the electrochemical concentration cell ozonesonde, Atmos. Meas. Tech, 11, 3661-3687, https://doi.org/10.5194/amt-11-3661-2018.

Tarasick, D.W., J. Davies, H.G.J. Smit and S.J. Oltmans (2016), A re-evaluated Canadian ozonesonde record: measurements of the vertical distribution of ozone over Canada from 1966 to 2013, Atmos. Meas. Tech. 9, 195-214, doi:10.5194/amt-9-195-2016.

Tarasick, D.W., H.G.J. Smit, A.M. Thompson G.A. Morris, J.C. Witte, J. Davies, T. Nakano, R. van Malderen, R.M. Stauffer, T. Deshler, B.J. Johnson, R. Stübi, S.J. Oltmans and H. Vömel (2019), Improving ECC Ozonesonde Data Quality: Assessment of Current Methods and Outstanding Issues, Earth and Space Science, in review.

Van Malderen, R., Allaart, M.A.F., De Backer, H., Smit, H.G.J., De Muer, D.: On instrumental errors and related correction strategies of ozonesondes: possible effect on

calculated ozone trends for the nearby sites Uccle and De Bilt, Atmos. Meas. Tech., 9, 3793-3816, doi:10.5194/amt-9-3793-2016, 2016.

Witte, J.C., A.M. Thompson, H.G.J. Smit, H. Vömel, F. Posny and R. Stübi (2018), First reprocessing of Southern Hemisphere ADditional OZonesondes profile records: 3. Uncertainty in ozone profile and total column. J. Geophys. Res., 123, 3243–3268. https://doi.org/10.1002/2017JD027791.

Witte, J.C., Thompson, A.M., Schmidlin, F.J., Northam, E.T., Wolff, K.R. and Brothers, G.B. (2019), The NASA Wallops Flight Facility digital ozonesonde record: Reprocessing, uncertainties, and dual launches. J. Geophys. Res.,124, 3565–3582. https://doi.org/10.1029/2018JD030098

---

## Author Comment (AC2) · 1 Nov 2019

Reply to Referee #2

Comment pg 4, lines 92-101: We agree. Text and references added.

Comment pg 4, line 97: Agree. Change made.

Comment pg 4, line 98: We agree that the stoichiometry is important, however it is not our intention to discuss the electro-chemistry of the ECC. Out purpose for showing data is to only demonstrate the potential capability of the digital bench. The list of uncertainties has been up-dated as suggested.

Comment pg 6, lines 159-167: The ECC-sensor flow measurements have been made with both automatic and bubble flow meter methods . . . MeteoSwiss made such tests with their digital bench and bubble flow meter a few years ago and found agreement to 1.1 percent . . . Similar data exists at Wallops with which we plan a statistical comparison, hopefully in time to add the results to the paper. We agree with the referee and have added the reference to Tarasick et al (2016).

Comment pg 8, line 230: We do not believe the use of 'hacking' is slang since the present use of the word 'hack' is now commonplace global wide. None the less, we have changed the sentence.

Comment pg 9, line 271: We have added. . . pressure and temperature at sea level and use of such calibrations at upper altitudes would be an ill-defined representation.

Comment pg, 10, lines 276-278: Good comment. We have cited Johnson et al (2002).

Comment pg 11, lines 325-325: We have referred to Johnson et al (2002) as suggested.

Comment pg 13, lines 369-370: We agree, the statement is argumentative and we have removed it. Similar comment was made by referee #1.

---

## Author Response (AR1)

Referee #1 Questions

General comments

The manuscript can be divided into two parts. In the first part the authors describe the design of a digital calibration bench for ECC ozonesondes in use since 2007 at the NASA/GSFC/Wallops Flight Facility. In the second part the digital calibration bench is used to test Science Pump Corp. 6A ECC ozonesondes with two different sensing solution types. In the first part the digital calibration bench itself is good described. Preparation of ozonesondes using such a device is superior to a manual preparation in particular when a UV photometer as a reference is used. This description alone qualifies for a publication in AMT. With respect to the second part it is not clear to me whether this part is only written to demonstrate the prospects of the bench as indicated e.g. at line 326 or to make valuable scientific statements. In a demonstration mode large portions, e.g. the BESOS discussion, can be omitted. For scientific statements the whole second part offers some potential for improvements, i.e. a better statistic and an error analysis. However, in total I recommend the publication of the manuscript after some minor revisions.

Specific comments

1. The title of the manuscript is dealing with the first part only. The title should address both parts in case the second part is not for demonstrations only.

2. As pointed out several times the aim of the digital calibration bench was to inves- tigate the behavior of ECC ozonesondes and to compare different configurations in a consistent and resources conserving manner replacing e.g. dual soundings. Al- though the advantage of reducing subjectivity compared to the manual preparation is mentioned, a clear statement is missing, that the bench is used at the Wallops Flight Facility for routine soundings (since when?), too. In this frame, one can address the fact that such calibration benches would be of benefit in particular for the ozonesonde records at remote sites with frequent exchange of operators (neglecting the needed financial effort).

3. Line 108: What means "similar" to the MeteoSwiss version? Are there improve- ments?

4. Line 159: Please list manufacturer, sensor type, measurement principle of the flow rate measurement device. The same is desired for the UV photometer.

5. Line 207: I am sure that the authors do know that the cathode and anode cells have to be filled in the right sequence and that the instructions are accordingly. Please give a small hint.

6. Line 233: "After recording 170 nb of ozone for one minute". Fig. 2 (upper panel) C2 tells "10 minutes" instead. I assume the 10 minutes are true.

7. Line 271: I suggest: "... bench is limited to pressure and temperature ranges appearing at sea level."

8. Lines 282-298: In order to classify some statements in this paragraph the statistical background, i.e. the number of investigated sondes, is needed already here. E.g. the background current can be batch dependent, which should relativize the statement at lines 291-293.

9. Lines 335: I would agree to substitute "ideal" by "good", since a negative aspect is mentioned right after.

10. As already mentioned before, the second part suffers from a missing statistical error analysis. Presented are only averaged data without error bars (or single cases). Without knowing the statistical errors it is impossible to justify whether the number of underlying cases is sufficient large.

11. Lines 341-342: Why is only one example shown here? For all other cases the averages were shown.

12. Lines 369-370: A first answer would be the final calibration. However, again, it would be helpful to see the other examples.

13. Lines 393-424: Is this (incl. Fig. 5) a new analysis not conducted in the BESOS publication before? BESOS outcomes had been already discussed at lines 330-333. However, a comparison to JOSIE2000 is missing. Why?

14. Lines 430-433: I disagree with the statement "... measured virtually the same ozone partial pressure until reaching 70-80 nb . . .". Obviously, the 0.5% sondes mea- sure significant less ozone in the lower troposphere, too. A plot showing the differences in relative units would be interesting.

15. A last comment for the future use: The test environment is bound to the surface C3

conditions. One might learn more how to use the bench calibrations within these limits by combining them with subsequent dual flights or chamber experiments like JOSIE.

Technical corrections

1. Line 45: Please use SI units throughout the manuscript, i.e. mPa instead of nb for the ozone partial pressure.

2. Line 49: Write out the acronym BESOS in the abstract, too. 3. Line 88: Delete one "the".

4. Line 250-252: The steps are in ozone partial pressure. In Fig. 2. upper panel the steps are given in mixing ratios. What is actually used?

5. Fig. 2 lower panel: - The blocks with "TEI Generator" and "Hi Ozone" seems to be misleading. As far as I understood the ozone is generated inside the generator and not outside. I guess the TEI Generator has one outlet, which sends Zero Air, when the generator inside is off, and Hi Ozone, when the generator is on. In that case V3 would be needless (or somehow hidden in the generator). Or, the generator has two outlets, one for Zero Air and one for Hi Ozone. In that case V3 makes sense. What is true? - If you use a different color for Hi Ozone please explain it in the legend. - The blue arrows at the barometer and the two current sensors indicate that the computer is triggering these devices. Is that right? - The writing of the word "Exhaust" near ECC Sensor P2 should be shifted to the right to the real exhaust. - How does the information of the mass flow measurement go into the computer? Is there a wired control connection (please indicate it in the diagram) or is it manually transferred by the operator (please note it in the main text)?

6. Fig. 3: Why does the plot differ somewhat from the first submitted version? Please comment in your reply only and not in the manuscript.

7. Fig. 6: Please add "N = 12" in the plot to be consistent with the other plots.

**Reply to Referee #1**

**Reply to General Comments**

*We acknowledge the referee's suggestion that this paper could be two parts. Our intention is to convey the idea of an automated bench and its usefulness. The data shown are examples meant to demonstrate results obtainable with the digital bench. We are removing the section discussing BESOS.*

**Reply to Specific comments**

Reply to specific comment #1

> *We intend to retain the present title since the examples given are meant to demonstrate the advantage of the bench.*

Reply to Specific comment #2.

> *We agree.  A statement will be included that addresses operational use of the bench. Note, the bench was used intermittently until 2017 when components began to fail and a resource to maintain the bench were not available.*

Reply to Specific Comment #3

> *There are no known improvements made to the Wallops bench although it is not as sophisticated as the MeteoSwiss unit. We are aware that the MeteoSwiss unit has been updated with up-to-date components.*

Reply to specific comment #4

> *Instrument information about the mass flow meter and UV photometer (TEI 49C) will be added.*

Reply to specific comment #5.

> *We have changed the text to indicate the sequence used to fill the cells.*

Reply to specific comment #6.

> *Text is wrong. Correction made, now reads 10 minutes*

Reply to specific comment #7.

*Agree. Text has been added.*

Reply to specific comment #8.

*Additional text will be added.*

Reply to specific comment #9.

*Agree. Replaced 'ideal' with 'useful'.*

Reply to specific comment #10.

*We are endeavoring to provide additional information. Figure 3 will be updated.*

Reply to specific comment #11.

*We believe one example is enough with which to describe the ECC characteristic discussed. One or two more such figures are possible, but we feel adds no additional information.*

Reply to specific comment #12.

*The sentence will be removed.*

Reply to specific comment #13.

*The BESOS discussion and Fig 5 are being removed.  JOSIE2000 is not discussed because there were no simultaneous measurements of SPC 6AECC's with 1.0 and 0.5 percent KI solutions prepared by the same lab. The ECC's also were prepared by different participating labs using that labs operational procedure.*

Reply to specific comment #14.

*We agree the statement could be argumentative and have removed it.*

Reply to specific comment #15.

*Unfortunately, dual flights using ECC's calibrated with the bench were not carried out.*

**Reply to Technical Comments**

Reply to technical comment #1.

*Changed nb to mPa.*

Reply to technical comment #2.

*Text and figures relating to BESOS have been removed.*

Reply to technical comment #3.

*Done. Removed the extra 'the'.*

Reply to technical comment #4.

*The use of ppb is an error and should be mPa.*

Reply to technical comment #5.

*There is one ozone generator outlet. HI OZONE is from an independent source. The computer prompt instructs the operator to turn HI OZONE on after which the computer handles the rest. The Figure is being corrected. There is a wired connection to the mass flow meter.*

Reply to technical comment #6

*The earlier plot was of a single measurement. Fig contains average measurements.*

Reply to technical comment #7.

*Will add the correct N=12.*

Referee # 2 Questions

This is a worthwhile paper, and should be published. I have a number of minor con- cerns that the authors may wish to address first, however.

Pg. 4, lines 92-101: Some mention of the efforts of the O3S-DQA initiative (Smit et al., 2012; Smit and ASOPOS panel, 2014) would be appropriate here. Perhaps even some of the recent re-evaluation papers (Tarasick et al., 2016; Van Malderen et al., 2016; Witte et al., 2018; 2019; Sterling et al., 2018) would not be out of place. The references Barnes (1982) and Barnes et al (1985) for sonde accuracy are rather old, and there are better ones, which the authors know as they co-authored some of them. There is a good summary in the forthcoming ASOPOS-2 report, also published as a paper in review for Earth and Space Science (Tarasick et al., 2019).

Pg. 4, line 97: "whether measured". Might insert "it is" to make comprehension easier for non-native speakers.

Pg. 4, line 98: "the use of the appropriate potassium iodide (KI) concentration". While the KI concentration does have an effect, the uncertainty really lies with the stoichiome- try of the KI reaction with ozone, as well as unwanted side reactions with the phosphate buffer. Losses of ozone and/or iodine in various ways should be included in this list, and motor speed might also be so included, since motors have changed in recent years.

Pg. 6, lines 159-167: What is the uncertainty of the automated flow rate measurement? This discussion seems to treat it as zero! The volumetric bubble flow method is quite accurate (and as a method traceable to physical constants, is typically used to calibrate automatic devices). Operator uncertainty is about 0.1-0.3% (Tarasick et al., 2016), less than 1/10 of what the authors suggest; the automated Gilibrator is only slightly better (if used properly).

Pg. 8, line 230: Insert "Measuring the. . ." before "Response". Line 242: "hacked" is slang; moreover it's not clear what is meant.

Pg. 9, line 271: Text missing here?
Pg. 10, lines 276-278: Should cite Johnson et al. (2002) here.

Pg. 11, lines 325-326: On the other hand, it's explained in great detail in Johnson et al. (2002). Why not refer to that?

Pg. 13, lines 369-370: Good question. The variation shown suggests a variability of about 5%, at least for the 0.5% solution. That is rather large, and serious investigation of it might add a lot to current understanding of ECC uncertainties, since, as the authors point out, such investigations are much easier to do than experiments at the World Ozone Calibration facility at Jülich.

*We agree, the statement is too argumentative and have removed it. Similar comment was made by referee #1.*

[revised manuscript text omitted]

Functional Diagram Ozonesonde Calibration Test Bench

*<object>*¶

¶

Fig 03.

[Figure]

Page Break

Fig 04.

[Figure]

[Figure]

Fig 05.

¶

[Figure]

.

Fig 06.

[Figure]

.

¶
Fig 07.

**Page 12: [1] Deleted**       **Schmidlin, Francis J. (WFF–610.W)[EMERITUS]**       **12/3/19 12:43:00 PM**

**Page 12: [1] Deleted**       **Schmidlin, Francis J. (WFF–610.W)[EMERITUS]**       **12/3/19 12:43:00 PM**

**Page 12: [1] Deleted**       **Schmidlin, Francis J. (WFF–610.W)[EMERITUS]**       **12/3/19 12:43:00 PM**

**Page 12: [1] Deleted**       **Schmidlin, Francis J. (WFF–610.W)[EMERITUS]**       **12/3/19 12:43:00 PM**

**Page 12: [1] Deleted**       **Schmidlin, Francis J. (WFF–610.W)[EMERITUS]**       **12/3/19 12:43:00 PM**

**Page 12: [1] Deleted**       **Schmidlin, Francis J. (WFF–610.W)[EMERITUS]**       **12/3/19 12:43:00 PM**

**Page 12: [1] Deleted**       **Schmidlin, Francis J. (WFF–610.W)[EMERITUS]**       **12/3/19 12:43:00 PM**

**Page 12: [1] Deleted**       **Schmidlin, Francis J. (WFF–610.W)[EMERITUS]**       **12/3/19 12:43:00 PM**

**Page 12: [1] Deleted**       **Schmidlin, Francis J. (WFF–610.W)[EMERITUS]**       **12/3/19 12:43:00 PM**

**Page 12: [2] Deleted**       **Schmidlin, Francis J. (WFF–610.W)[EMERITUS]**       **11/29/19 1:49:00 PM**

**Page 12: [2] Deleted**       **Schmidlin, Francis J. (WFF–610.W)[EMERITUS]**       **11/29/19 1:49:00 PM**

**Page 12: [2] Deleted**       **Schmidlin, Francis J. (WFF–610.W)[EMERITUS]**       **11/29/19 1:49:00 PM**

**Page 12: [2] Deleted**       **Schmidlin, Francis J. (WFF–610.W)[EMERITUS]**       **11/29/19 1:49:00 PM**

**Page 12: [2] Deleted**       **Schmidlin, Francis J. (WFF–610.W)[EMERITUS]**       **11/29/19 1:49:00 PM**

**Page 12: [2] Deleted**  Schmidlin, Francis J. (WFF–610.W)[EMERITUS]  11/29/19 1:49:00 PM

**Page 12: [2] Deleted**  Schmidlin, Francis J. (WFF–610.W)[EMERITUS]  11/29/19 1:49:00 PM

**Page 12: [2] Deleted**  Schmidlin, Francis J. (WFF–610.W)[EMERITUS]  11/29/19 1:49:00 PM

**Page 12: [2] Deleted**  Schmidlin, Francis J. (WFF–610.W)[EMERITUS]  11/29/19 1:49:00 PM

**Page 12: [2] Deleted**  Schmidlin, Francis J. (WFF–610.W)[EMERITUS]  11/29/19 1:49:00 PM

**Page 12: [2] Deleted**  Schmidlin, Francis J. (WFF–610.W)[EMERITUS]  11/29/19 1:49:00 PM

**Page 12: [2] Deleted**  Schmidlin, Francis J. (WFF–610.W)[EMERITUS]  11/29/19 1:49:00 PM

**Page 12: [2] Deleted**  Schmidlin, Francis J. (WFF–610.W)[EMERITUS]  11/29/19 1:49:00 PM

**Page 12: [2] Deleted**  Schmidlin, Francis J. (WFF–610.W)[EMERITUS]  11/29/19 1:49:00 PM

**Page 12: [2] Deleted**  Schmidlin, Francis J. (WFF–610.W)[EMERITUS]  11/29/19 1:49:00 PM

**Page 12: [2] Deleted**  Schmidlin, Francis J. (WFF–610.W)[EMERITUS]  11/29/19 1:49:00 PM

**Page 12: [2] Deleted**  Schmidlin, Francis J. (WFF–610.W)[EMERITUS]  11/29/19 1:49:00 PM

**Page 12: [2] Deleted**  Schmidlin, Francis J. (WFF–610.W)[EMERITUS]  11/29/19 1:49:00 PM

**Page 12: [2] Deleted**  Schmidlin, Francis J. (WFF–610.W)[EMERITUS]  11/29/19 1:49:00 PM

**Page 12: [2] Deleted**      **Schmidlin, Francis J. (WFF–610.W)[EMERITUS]**      **11/29/19 1:49:00 PM**

**Page 12: [2] Deleted**      **Schmidlin, Francis J. (WFF–610.W)[EMERITUS]**      **11/29/19 1:49:00 PM**

**Page 12: [2] Deleted**      **Schmidlin, Francis J. (WFF–610.W)[EMERITUS]**      **11/29/19 1:49:00 PM**

**Page 12: [2] Deleted**      **Schmidlin, Francis J. (WFF–610.W)[EMERITUS]**      **11/29/19 1:49:00 PM**

**Page 12: [2] Deleted**      **Schmidlin, Francis J. (WFF–610.W)[EMERITUS]**      **11/29/19 1:49:00 PM**

**Page 12: [2] Deleted**      **Schmidlin, Francis J. (WFF–610.W)[EMERITUS]**      **11/29/19 1:49:00 PM**

**Page 12: [2] Deleted**      **Schmidlin, Francis J. (WFF–610.W)[EMERITUS]**      **11/29/19 1:49:00 PM**

**Page 12: [2] Deleted**      **Schmidlin, Francis J. (WFF–610.W)[EMERITUS]**      **11/29/19 1:49:00 PM**

**Page 12: [2] Deleted**      **Schmidlin, Francis J. (WFF–610.W)[EMERITUS]**      **11/29/19 1:49:00 PM**

**Page 12: [2] Deleted**      **Schmidlin, Francis J. (WFF–610.W)[EMERITUS]**      **11/29/19 1:49:00 PM**

**Page 12: [2] Deleted**      **Schmidlin, Francis J. (WFF–610.W)[EMERITUS]**      **11/29/19 1:49:00 PM**

**Page 12: [2] Deleted**      **Schmidlin, Francis J. (WFF–610.W)[EMERITUS]**      **11/29/19 1:49:00 PM**

**Page 12: [2] Deleted**      **Schmidlin, Francis J. (WFF–610.W)[EMERITUS]**      **11/29/19 1:49:00 PM**

**Page 12: [2] Deleted**      **Schmidlin, Francis J. (WFF–610.W)[EMERITUS]**      **11/29/19 1:49:00 PM**

**Page 12: [2] Deleted**      **Schmidlin, Francis J. (WFF–610.W)[EMERITUS]**      **11/29/19 1:49:00 PM**

**Page 12: [2] Deleted**      **Schmidlin, Francis J. (WFF–610.W)[EMERITUS]**      **11/29/19 1:49:00 PM**

**Page 12: [2] Deleted**      **Schmidlin, Francis J. (WFF–610.W)[EMERITUS]**      **11/29/19 1:49:00 PM**

**Page 12: [2] Deleted**      **Schmidlin, Francis J. (WFF–610.W)[EMERITUS]**      **11/29/19 1:49:00 PM**

**Page 12: [2] Deleted**      **Schmidlin, Francis J. (WFF–610.W)[EMERITUS]**      **11/29/19 1:49:00 PM**

**Page 12: [2] Deleted**      **Schmidlin, Francis J. (WFF–610.W)[EMERITUS]**      **11/29/19 1:49:00 PM**

**Page 12: [2] Deleted**      **Schmidlin, Francis J. (WFF–610.W)[EMERITUS]**      **11/29/19 1:49:00 PM**

**Page 12: [2] Deleted**      **Schmidlin, Francis J. (WFF–610.W)[EMERITUS]**      **11/29/19 1:49:00 PM**

**Page 12: [2] Deleted**      **Schmidlin, Francis J. (WFF–610.W)[EMERITUS]**      **11/29/19 1:49:00 PM**

**Page 12: [2] Deleted**      **Schmidlin, Francis J. (WFF–610.W)[EMERITUS]**      **11/29/19 1:49:00 PM**

**Page 12: [2] Deleted**      **Schmidlin, Francis J. (WFF–610.W)[EMERITUS]**      **11/29/19 1:49:00 PM**

**Page 12: [2] Deleted**      **Schmidlin, Francis J. (WFF–610.W)[EMERITUS]**      **11/29/19 1:49:00 PM**

**Page 12: [2] Deleted**      **Schmidlin, Francis J. (WFF–610.W)[EMERITUS]**      **11/29/19 1:49:00 PM**

**Page 12: [2] Deleted**      **Schmidlin, Francis J. (WFF–610.W)[EMERITUS]**      **11/29/19 1:49:00 PM**

**Page 12: [2] Deleted**      **Schmidlin, Francis J. (WFF–610.W)[EMERITUS]**      **11/29/19 1:49:00 PM**

**Page 12: [2] Deleted**       Schmidlin, Francis J. (WFF−610.W)[EMERITUS]       11/29/19 1:49:00 PM

**Page 12: [2] Deleted**       Schmidlin, Francis J. (WFF−610.W)[EMERITUS]       11/29/19 1:49:00 PM

**Page 12: [2] Deleted**       Schmidlin, Francis J. (WFF−610.W)[EMERITUS]       11/29/19 1:49:00 PM

**Page 12: [2] Deleted**       Schmidlin, Francis J. (WFF−610.W)[EMERITUS]       11/29/19 1:49:00 PM

**Page 15: [3] Deleted**       Schmidlin, Francis J. (WFF−610.W)[EMERITUS]       11/12/19 11:50:00 AM

**Page 15: [4] Deleted**       Schmidlin, Francis J. (WFF−610.W)[EMERITUS]       11/12/19 11:53:00 AM

**Page 15: [5] Deleted**       Schmidlin, Francis J. (WFF−610.W)[EMERITUS]       11/29/19 2:51:00 PM

---

## Author Response (AR2)

**Associate Editor Decision: Publish subject to minor revisions (review by editor) (14 Jan 2020) by Roeland Van Malderen Comments to the Author: Dear authors, Thank you for taking most of the reviewer comments into account. However, some minor revisions and clarifications are still needed.**

Especially the discussion of Fig 4 needs to be generalized. As also brought up by the first reviewer, this figure and its discussion only deals with one example of two ECCs, tested at three different weeks. However, in lines 380-381, you wrote that "a number of time-separated calibrations were conducted". So, how many of these experiments have been conducted? And why do you not present the means of those experiments (i.e. average values at 0, 5, 10, 15, 20, 25, 30 mPa of those experiments) at the different weeks, as in Fig. 3, instead of showing just one example? Is the shown example representative for the other experiments as well? Please comment. Furthermore, the discussion focuses on the calibration for 30 mPa, which is, an unrealistic high ozone amount for the stratosphere. Can you also be conclusive for the finding that "the 30 mPa response of the ECCs increases with the week (compared to the reference)" for other ozone partial pressures? Please be more specific and more general.

In this context, your suggestion in lines 404-405 that "On the other hand, the changes could simply be a normal evolution of typical performance behavior" only holds if the illustrated performance is consistent (i) for the different time-separated calibration tested ozonesondes and (ii) for different ozone partial pressure levels (not only for 30 mPa). The same argument holds for the conclusion of this test in the summary section, lines 491-493 ("Results from testing ECC cells over a period of three weeks, one test each week, showed the calibration changed, e.g. about 10 percent for 1.0 percent KI and about 4-5 percent for the 0.5 percent solutions."): it is not clear at all if this conclusion is valid only for the shown example and the numbers are true for all ozone partial pressure levels!

REPLY: We agree that the single example given in Figure 4 is insufficient to suggest ECC calibration increases weekly over a three-week period. We have reviewed the available sets (11) of 'three-weekly' calibrations and found some calibrations did increase but in the average of these data found small week-to-week changes for both the 1.0 and 0.5 percent KI solutions, but these were very small.

Because of the nebulous nature of these results we opted to remove Figure 4 and the applicable text.

Line 44: already mention in the abstract that this study only deals with Science Pump Corp. ECCs

REPLY: Because a one reviewer raised the question to be more specific as to the ECC manufacturer, we prefer for clarity, to maintain the present reference to SPC in the introduction.

Line 72: write out EnSci here.

REPLY: Environmental Science (EnSci) as been added to the text where it is first mentioned. See Section 2.1 .

Lines 103-104: As asked by the second reviewer: use a more up-to-date reference for the accuracy of ozonesonde measurements here. E.g. the WMO GAW Report 201 (Smit & ASOPOS Panel 2014) gives an estimate (perhaps also Deshler et al., 2017; Thompson et al., BAMS, 2019).

REPLY: As suggested, text was changed indicating reference change.

Lines 119-130: provide the years at which the digital calibration benches were/are operationally used at Wallops, Payerne and Nairobi

REPLY: Text added showing Payerne since 1995, Nairobi since 2018, Wallops Island (development 2005-2008; operational 2009-2017-only used for preparation, no calibration).

Line 139: Write out TEI here.

REPLY: Added Thermo Environmental Instruments (TEI) where it first appears in Section 2.1

Lines 240-243: please check the order of filling the cells. If at Wallops, like you wrote it down, the anode cells are filled before the cathode cells, the SOPs are not followed, which is a major issue for the conclusions reached.

REPLY: Text has been corrected.

Lines 288-289: "Generally, the downward calibration experiences small differences from the upward calibration". In which sense? Consistent for all measurements? Related to the sensor response and consequently its memory of higher vs. lower ozone amounts?

REPLY: We have added text indicating that the measured partial pressure during the downward calibration is consistently higher for both 1.0 and 0.5 percent KI solutions. Apparently, the ECC sensor retains the memory of experiencing high ozone concentration.

Lines 325-327: "The final background currents often were somewhat higher than background currents experienced with manual preparation, generally about 0.04 microns" Please specify which final background current you mean here (final background current obtained prior to balloon release??) and give a possible reason why these background currents are higher with the automated procedure than with the manual procedure. Is this because of the linear calibration step in the automated procedure?

REPLY: Preparation with the digital bench obtains the final background current after experiencing the high ozone concentration of the calibration step (0-30 mPa). We expect that the residual memory of the ECC sensor is the reason for background currents higher than the manual. The text has been changed to reflect this effect. The manual value is obtained just prior to balloon release.

Section 3.2: you should mention in the beginning of this section that the 1.0% 1.0 B solution strength is the recommended one for Science Pump (e.g. Smit & ASOPOS, WMO GAW report 201, Deshler et al., 2017).

REPLY: Text added reflecting this.

Lines 365-367: Please repeat here the explanation given by Johnson et al. (2002) about the effect of different KI solution concentrations and the side effects from the buffers used, within the context of your 1.0%1.0B & 0.5%0.5B SPC comparison.

REPLY: Text has been added reflecting explaination from Johnson et al, (2002).

Line 436: The profiles were averageD

REPLY: Fixed. Thank you.

Lines 436-443: what about the total ozone normalization factors of the 12 dual flights? Which solution strengths are closer to the co-located Brewer/Dobson or satellite overpass total ozone measurements?

REPLY: Normalization is not done at Wallops Island. Dobson total ozone compares very well with the 0.5 percent KI solution. ECC 1.0 and 0.5 percent KI total ozone vs. Dobson total ozone has been added to the paper, e.g., the sample of 12 profil- mean DU for the ECC 1.0 % is 330.4 DU, ECC 0.5 % 308.3 DU, and 309.5 DU for the Dobson.

Lines 456-460: what about the total ozone normalization factors of the three dual flights? Which solution strengths are closer to the co-located Brewer/Dobson or satellite overpass total ozone measurements?

REPLY: Reference to dual-flights was removed. The authors considered the amount of information too slim.

Fig 04: I guess the units in these figures should be ranging from 0 to 30 mPa instead of 0 to 300?

REPLY: Figure 4 was removed from paper.

Fig 06: I guess the legend should show the 0.3% KI in red, instead of 0.5% KI. As in Fig. 3, you might also list the standard deviations

REPLY: Correction has been made and the std dev added. New figure added.

[revised manuscript text omitted]

Functional Diagram Ozonesonde Calibration Test Bench

[Figure]

Fig 03.

[Figure]

¶

Fig 04.

[Figure]

[Figure]

Page Break

Fig 05.

[Figure]

---

## Author Response (AR3)

**Associate Editor Decision: Publish subject to technical corrections** (27 Jan 2020) by Roeland Van Malderen

Comments to the Author:

* l 139: analyzer instead of analyzeer

*Corrected.*

* l 141: illustrateS

Corrected.

* l 253-254: you mentioned in your response to me why the downward calibrations are always higher than the upward calibrations. Please include this argument also in the text.

*Argument as requested has been included.*

* l 381: Dobson measurements at Wallops, I assume? Please specify and you might also give a reference to a paper where the Dobson dataset at Wallops is presented.

*Dobson data have been available since 1963. One references added.*

* l 384: 20.9 DU instead of 20./9 DU

*Corrected.*

* Caption Fig. 5: 0.3% 0.3B instead of 0.3% 0.5B

*Corrected.*

* Fig 5: change the labels "1.0% mPa" and "0.3% mPa" for the blue and red curves (use % instead of mPa)!

*Correction made to Figure.*

[revised manuscript text omitted]

**Manual insertion of KI solution required**

Fig 02.

**ECC Calibration System Sequential Flow Diagram**

[Figure]

10/29/19 etn

Functional Diagram Ozonesonde Calibration Test Bench

[Figure]

10/28/19 etn (from 6/7/05 TB)

Fig 03.

[Figure]

Fig 04.

[Figure]

Fig 05.

[Figure]

[Figure]